# Septin Organization and Dynamics for Budding Yeast Cytokinesis

**DOI:** 10.3390/jof10090642

**Published:** 2024-09-09

**Authors:** Maritzaida Varela Salgado, Simonetta Piatti

**Affiliations:** CRBM (Centre de Recherche en Biologie cellulaire de Montpellier), University of Montpellier, CNRS UMR 5237, 34293 Montpellier, France; maritzaida.varela@ijm.fr

**Keywords:** *S. cerevisiae*, septins, cytokinesis

## Abstract

Cytokinesis, the process by which the cytoplasm divides to generate two daughter cells after mitosis, is a crucial stage of the cell cycle. Successful cytokinesis must be coordinated with chromosome segregation and requires the fine orchestration of several processes, such as constriction of the actomyosin ring, membrane reorganization, and, in fungi, cell wall deposition. In *Saccharomyces cerevisiae*, commonly known as budding yeast, septins play a pivotal role in the control of cytokinesis by assisting the assembly of the cytokinetic machinery at the division site and controlling its activity. Yeast septins form a collar at the division site that undergoes major dynamic transitions during the cell cycle. This review discusses the functions of septins in yeast cytokinesis, their regulation and the implications of their dynamic remodelling for cell division.

## 1. Introduction

Septins form an evolutionarily conserved family of cytoskeletal proteins expressed in many eukaryotic organisms but absent in land plants [1,2]. Septins were initially discovered in budding yeast through mutagenesis screens aimed at identifying genes involved in the cell division cycle (*CDC* [3]). Temperature-sensitive mutations affecting the genes *CDC3*, *CDC10*, *CDC11*, and *CDC12* cause hyperpolarized growth and cytokinesis failure at high temperatures, suggesting a role in cell morphogenesis and cell division [4,5]. Electron and immunofluorescence microscopy studies revealed that the products of these genes localize at the bud neck, forming a filamentous collar [6,7,8]. Given their role in septation, these proteins were named “septins”. Subsequently, three additional septin genes were identified in budding yeast based on sequence homology: *SHS1*, expressed like the aforementioned septins in vegetative cells, and *SPR3* and *SPR28*, which are expressed exclusively during meiosis [9,10,11]. 

Yeast septins participate in a wide range of cellular functions, including cell polarity [12], cell cycle progression [13,14,15], spindle positioning [16], cytokinesis and septum formation [4,17,18,19]. Since their discovery in budding yeast, septin orthologs have been identified in animals, fungi and protists, where they are involved in numerous cellular processes, such as cytokinesis [20,21,22,23,24,25], exocytosis [26], phagocytosis [27], morphogenesis [28,29,30,31,32], ciliogenesis [33,34,35], cell motility [36,37], spermatogenesis [38], lipid metabolism [39,40], synaptic activity [41,42,43] and bacterial entrapment during cell infection [44,45].

Given their pivotal roles in cellular organization and function, it is not surprising that septins are implicated in a range of pathological conditions. Indeed, septin dysfunction has been associated with cancer, neurodegenerative disorders, infectious diseases and infertility (reviewed in [46]).

Septins are GTP-binding proteins characterized by a conserved GTPase domain and variable N-terminal and C-terminal regions. A septin-unique element is a distinctive feature of all septins, although its precise function remains unknown [47]. Septins invariably assemble into rod-shaped hetero-complexes composed of two copies of each septin monomer organized in a palindromic fashion (reviewed in [48]). Septin hetero-complexes then interact to form non-polar filaments and/or higher-order structures, such as bundles, rings or gauzes, as observed both in vitro and in vivo across various organisms [6,22,24,31,38,49,50,51,52,53,54,55,56,57,58,59,60]. 

Membranes play a crucial role in septin organization. Septins associate with lipid membranes containing negatively charged phospholipids, particularly phosphatidylinositol 4,5-bisphosphate (PI(4,5)P_2_) [61,62,63,64,65]. In budding yeast, PI(4,5)P_2_ is enriched at the presumptive bud site in G1 [66], thus accounting for septin recruitment at this specific cellular location. Additionally, perturbing the amount of PI(4,5)P_2_ at the plasma membrane causes the septin collar to be disassembled at the bud neck and/or septin filaments to form aberrant structures cytoplasm, such as small rings and arcs [64,67], indicating that PI(4,5)P_2_ is crucial for septin organization in vivo. 

How septins bind to membranes is unclear. A protein motif rich in basic residues at the N-terminus of septins has been implicated in membrane binding via electrostatic interactions with the negative charges of phospholipids [63,68,69,70]. However, whether this polybasic motif is actually exposed on the surface of septins or buried, thus precluding a prominent role in membrane binding, is still debated [48]. 

Septin binding to membranes is also affected by membrane curvature, showing a strong preference for micrometric curvatures [71,72,73]. Consequently, septins are frequently located at curved membranes, such as the hyphal branch sites of filamentous fungi, the base of primary cilia, and the annulus of sperm tails [31,32,33,50,74,75].

Finally, multiple septin-associated proteins and post-translational modifications are likely involved in septin organization in different physiological contexts, highlighting the complexity and dynamic nature of septin regulation.

## 2. Budding Yeast Septins and Their Localization

In vegetative cells, budding yeast septins form two hetero-octamers with a common core composed of Cdc12-Cdc3-Cdc10-Cdc10-Cdc3-Cdc12 and comprising either Cdc11 or Shs1 at both extremities [58,59,76]. Surprisingly, septin octamers always have symmetric termini, while asymmetric complexes capped by Cdc11 and Shs1 at each end cannot be formed, at least in vitro [76]. The linear septin complexes are often referred to as septin “rods” and measure 32–35 nm in length and 4–5 nm in diameter [58,59]. In vitro reconstitution assays using recombinant septins have shown a striking difference between Cdc11-capped and Shs1-capped yeast octamers, where the former can spontaneously polymerize end-to-end to form long, paired filaments in solution, while the latter appear unable to polymerize [50,58,59,77,78]. Shs1-capped octamers can nevertheless interact laterally and stagger on top of each other to form curved bundles, rings and spirals in vitro [59]. Additionally, in the presence of lipids or upon an *SHS1* mutation mimicking constitutive phosphorylation of Ser259, Shs1-capped octamers form gauze-like structures in addition to rings, suggesting that the cellular context and post-translational modifications influence the supramolecular organization of septins [59,60]. Despite being unable to polymerize into linear filaments, Shs1-capped rods can form heterotypic interactions with the termini of Cdc11-capped octamers [78], suggesting that they may be interspersed in septin filaments and higher-order structures. Consistently, the five mitotic septins display a completely superimposable localization in vegetative cells throughout the cell cycle. 

Formation of septin filaments is essential for cell division, as shown by the lethality of mutants incapable of septin polymerization [79]. This may explain why Cdc11 is required for viability in most yeast backgrounds, while Shs1 is dispensable [10,47,79]. During sporulation, the aforementioned septin hetero-octamers are replaced by the Spr28-Spr3-Cdc3-Cdc10-Cdc10-Cdc3-Spr3-Spr28 meiotic-specific complex, where Spr28 and Spr3 replace the terminal septin and Cdc12, respectively [80]. Unlike its mitotic Cdc11-capped counterpart, the Spr28-capped meiotic complex is unable to polymerize in solution, but it readily forms filaments on PI(4,5)P_2_-containing lipid monolayers [80], suggesting that its polymerization requires binding to the plasma membrane. Its function during meiosis is linked to proper biogenesis of the prospore membrane and cell wall assembly, thereby ensuring high sporulation efficiency [81]. 

The functional organization of septins is based on their assembly into higher-ordered structures. In vegetative cells, septins undergo dynamic changes in their structural organization during the cell cycle. First, in late G1, septins form a dynamic cortical ring at the presumptive bud site. Then, as the bud emerges, the septin ring expands into an hourglass-shaped structure, referred to as septin collar, around the bud neck. Finally, at the onset of cytokinesis, the septin collar splits into a double ring that sandwiches the AMR (Figure 1). This dramatic septin remodelling, which is essential for cell division (see below), is often referred to as septin ring splitting. Each split ring then persists in the mother and daughter cell, respectively, until the next cell cycle [7,82,83]. Florescence recovery after photobleaching (FRAP) experiments have shown that the newly forming septin ring is dynamic, as shown by its ability to exchange septins from an unassembled pool. In contrast, the septin collar is a stable and immobile structure. At cytokinesis, split septin rings become again relatively dynamic [84,85]. 

In cells exposed to mating pheromones, septins form a fuzzy band or a set of parallel bars at the base of the polarized projection (i.e. the shmoo) [7,83,86]. Upon cell fusion during mating, septins form an annulus at the midzone of the zygote that seems to affect the redistribution of supramolecular complexes and organelles [87]. 

During meiosis I, septins first form ring-like structures at the leading edge of membrane sacs, also known as prospore membranes, that originate in close apposition to the cytoplasmic face of the spindle pole bodies and that will later extend to engulf each of the four haploid nuclei. At the end of meiosis II, septins seem to localize quite uniformly at the plasma membrane surrounding the developing spore [9,88] (Figure 2). Interestingly, the septin Cdc10 accumulates in each mature spore to form a cortical cluster opposite to its neighboring sister spores in the ascus. This septin cluster serves as a prepolarity marker to direct the later polarised growth to penetrate the ascus wall during germination [89,90] (Figure 2). Additionally, upon germination but before bud emergence, Cdc10 localises as a band at the boundary between the two unequal halves of the germinating spore, suggesting that septins may form a cortical barrier between the growing and non-growing parts of the germinating spore [90].

Thus, budding yeast septins display a wide range of high order architectures depending on the physiological context, but they seem to be invariably associated with membranes, while in other eukaryotes, septins clearly also associate with the actin and microtubule cytoskeleton (reviewed in [91]). 

## 3. Septin Organization and Cytokinesis in Budding Yeast

Our current knowledge of septin architecture at the bud neck of mitotic yeast cells comes mainly from ultrastructural studies. Freeze-fracture and platinum-replica electron microscopy on unroofed spheroplasts or cryo-electron tomography on intact cells have shown that the mature septin collar at the bud neck is made by a network of axial filaments lying parallel to the mother bud axis and circumferential filaments that are orthogonally oriented, thus forming a gauze-like meshwork (Figure 3) [49,52,92]. Conversely, the septin double ring at cytokinesis is made exclusively of circumferential filaments [92], in line with polarized fluorescence microscopy studies showing that the average orientation of septin filaments in the split septin rings is perpendicular relative to the polarity axis [93,94,95]. Since fluorescently labelled septins show a marked decrease in fluorescence at the bud neck at the moment of septin ring splitting [18,85,94,96], current models envision that at the onset of cytokinesis the axial septin filaments disassemble and possibly partially reassemble into two arrays of circumferential filaments, forming the double ring (reviewed in [97]).

Cytokinesis is a delicate process that must precisely partition an equal complement of the replicated genomes and organelles to each daughter cell. It relies on assembly and contraction of an actomyosin ring (AMR) and must be tightly controlled and coordinated with other cell cycle processes, such as chromosome segregation. Typically, yeast cytokinesis can be divided into four steps: selection of the division site, assembly of the AMR, cleavage furrow ingression and primary septum formation powered by the AMR, and localized membrane remodelling. In budding yeast, septins are involved in most of these cytokinetic steps by adopting different structural organizations. 

The first steps of cytokinesis occur very early during the cell cycle. The selection of the bud site, which is also the division site, is determined in late G1 by the cell polarization machinery controlled by the master polarity GTPase Cdc42 that also recruits septins (reviewed in [98]; see below). Bud site selection follows a very specific pattern depending on whether cells are haploid or diploid. In haploid cells, the emergence of a new bud occurs proximally to the division site (axial budding pattern), while in diploid cells, it occurs distally (bipolar budding pattern). Such a stereotypical pattern depends on polarity landmark proteins that precisely mark the future bud site (reviewed in [98]). Despite their recruitment before bud emergence, septins are not required for bud emergence [4,99], but they contribute to the axial budding pattern by recruiting to the bud neck landmark proteins, such as Bud3 and Bud4 [100,101,102,103]. In addition, they facilitate the formation of a narrow cluster of Cdc42 at the presumptive bud site by recruiting GTPase-activating proteins (GAPs) that in turn inhibit Cdc42 in the membrane area surrounding its activated cluster (Figure 3) [104]. 

Before bud emergence, Myo1, the sole myosin-II heavy chain in budding yeast, is recruited along with its regulatory light-chain Mlc2 in a septin-dependent manner, forming a ring (Figure 1) [19,105,106,107]. At the time of the bud emergence, the septin ring expands around the bud neck, forming an hourglass-shaped structure that remains stable during mitosis. AMR assembly proceeds until the end of anaphase (Figure 1), when the IQGAP Iqg1 appears at the bud neck to recruit actin [19,105,108,109]. Around the same time, the essential myosin light chain Mlc1 arrives at the bud neck to interact with Myo1 and to enhance Myo1 targeting to the neck, thus further increasing its local levels [106,110]. Concomitantly, Mlc1 also promotes Iqg1 recruitment to the bud neck by binding to its IQ motifs [111]. Two formins, Bni1 and Bnr1, are redundantly essential for actin polymerization at the AMR [112,113]. Furthermore, formins have been shown to contribute to the accumulation of Mlc1 at the bud neck during cytokinesis [114], suggesting an additional mechanism by which they could participate in AMR assembly. After mitotic exit, the chitin synthase Chs2 is recruited to the AMR to promote formation of the primary septum along with AMR constriction [115,116,117,118,119]. Additionally, a protein complex made by Hof1, Inn1 and Cyk3 also joins the AMR and partially constricts alongside it to boost Chs2 recruitment and activity [113,116,120,121,122,123,124,125,126,127,128]. Several of the aforementioned AMR components and regulators, namely Myo1 [19,105], Mlc1 [106,108], Bnr1 [129,130] and Hof1 [128], require septins for their bud neck localization, suggesting that the septin collar orchestrates AMR assembly. 

Constriction of the AMR marks the beginning of cytokinesis (Figure 1). AMR constriction may occur partly through sliding of bipolar myosin filaments along actin filaments that are tethered to the plasma membrane, similar to the way by which actomyosin generates force in the striated muscle [131,132]. However, actin depolymerization by cofilin plays a predominant role in budding yeast AMR constriction [133]. Iqg1 degradation by the APC/C complex might also contribute to AMR disassembly [134,135].

AMR constriction drives membrane invagination inward and is also coupled to targeted deposition of post-Golgi vesicles to the division site to increase surface area and deliver the chitin synthase-II Chs2 to drive primary septum formation (PS) [118,136,137]. Thus, AMR constriction guides PS formation, which in turn stabilises the AMR during constriction [138,139,140]. 

At the onset of cytokinesis, the septin hourglass is split into two rings that sandwich the AMR (Figure 1) [82,96,141]. This septin reorganization is an essential prerequisite for cytokinesis [96,142]. Indeed, lack of septin ring splitting prevents AMR constriction and PS formation. Thus, the septin collar has both a positive and a negative role in cytokinesis: during mitosis, it organises the cytokinetic machinery by recruiting several cytokinetic proteins to the bud neck, but at cytokinesis, it must be displaced from the bud neck (through septin ring splitting or clearance altogether) to allow AMR constriction [96,142]. Consistently, septins are not required for AMR constriction once the AMR has been assembled [96,143]. The septin double ring could nevertheless facilitate plasma membrane closure by acting as a diffusion barrier that concentrates membrane remodelling factors, such as the polarisome and the exocyst complex, to the cleavage site [143]. 

Once the primary septum has been laid down, a secondary septum (SS) is assembled by glucan synthases and the chitin synthase Chs3 on both the mother and daughter sides of the PS, thus leading to abscission (Figure 1) [125,136]. By recruiting Hof1 to the bud neck, which in turn inhibits precocious activation of Chs3, septins indirectly set the timing of SS assembly so that it occurs only after the PS has been deposited [121,144]. 

The final step of yeast cytokinesis is cell separation (Figure 1), which involves PS digestion and SS remodeling by chitinase and endoglucanases, respectively (reviewed in [145]). If and how septins are involved in the control of cell separation is not known. Intriguingly, however, in fission yeast glucanases are localized in between the two split septin rings and are unable to form a proper ring at the cortex in septin mutants [146], suggesting that the septin double ring may form a diffusion barrier for the enzymes involved in cell separation.

## 4. The Control of Septin Dynamics during the Cell Cycle

We can distinguish three main steps in the dynamics of septin architecture during the budding yeast cell cycle: (1) septin recruitment and ring assembly; (2) maturation into the septin collar; and (3) transition into the septin double ring. 

### 4.1. Septin Recruitment and Ring Assembly 

Septins accumulate at the presumptive bud site initially as a cloud or patch (Figure 3). This process depends on the Rho GTPase Cdc42, which is a conserved master regulator of cell polarity [82,84,147]. Indeed, Cdc42 depletion leads to an unbudded cell phenotype lacking cortical septin structures [148], and *cdc42* mutants are defective in septin recruitment to the presumptive bud site or display septin mislocalization [103,147,149]. However, in some of these mutants Cdc42 cannot hydrolyze GTP and is locked in a GTP-bound state, while overexpression of Cdc42 GTPase-activating proteins (GAPs) can restore proper septin localization and function in septin mutants [147,148]. These observations have led to the proposal that Cdc42 must continuously cycle between a GTP- and a GDP-bound state in order to gather septins at the future bud neck [149]. Alternatively, active GTP-bound Cdc42 may be required to establish the first polarity cue necessary for septin recruitment (e.g., to define a membrane domain permissive for septin accumulation), while the active GTPase actually prevents septins from piling up. 

Once at the plasma membrane, septins rapidly organise into a cortical ring (Figure 3). Polarized exocytosis and insertion of membrane vesicles by Cdc42 and the exocyst complex are critical to this process by inhibiting septin accumulation in the center of the septin cap at the incipient bud site, thus creating the ring [104]. Thus, it is possible that the septin mislocalization phenotype of some GTP-bound *cdc42* mutants is accounted for by excessive or untargeted membrane vesicles tethering that disrupts septin ring assembly.

Several studies have attempted to link the function of Cdc42 in septin ring assembly to specific Cdc42 effectors (Figure 3). The p21-activated kinases (PAK) Cla4 and Ste20 are turned on by Cdc42 and Cla4 has been implicated in septin recruitment to the presumptive bud site through phosphorylation of several septins [99,150,151,152]. On the other hand, Cla4 binds the Cdc42 membrane scaffold Bem1 and recruits and phosphorylates Cdc24, the GEF (guanine-nucleotide exchange factor) for Cdc42, thus promoting its GTP-bound state [153,154,155]. Such a positive feedback loop suggests that the role of PAK kinases in septin recruitment may be partly mediated by a global enhancement in Cdc42 activity. In parallel, the Cdc24-Bem1 complex binds Cdc11, helping to bring septins to the bud site [156]. Two additional Cdc42 effectors, i.e. the paralogous membrane proteins Gic1 and Gic2, interact directly with septins and are involved in septin recruitment and organization in vivo and in vitro [104,148,157,158,159]. Accordingly, septin deposition and budding mostly fail in *gic1 gic2* double mutants at high temperatures [148]. Gic1 colocalizes with septins at the presumptive bud site at early stages of the cell cycle and at the bud neck later on [148], and it has been shown to bundle and cross-link septin filaments in vitro, thereby stabilizing them [159]. However, similar to PAK kinases, Gic1 and Gic2 are also involved in a positive feedback loop for Cdc42 activation, and overexpression of *CDC42* suppresses the lethality of cells lacking Gic1 and Gic2 at high temperatures [160,161], suggesting that septin recruitment to the future bud site may be empowered by Cdc42 itself or by effectors other than Gic proteins. 

PAK kinases possess a pleckstrin-homology domain that binds preferentially to PI(4,5)P_2_ [162]. Similarly, Gic2 (and possibly Gic1) bears a cluster of basic residues that binds to PI(4,5)P_2_ [158]. Thus, the PI(4,5)P_2_ binding properties of these Cdc42 effectors might influence their interaction with and organization of septin into a ring. 

The formin Bni1 contributes to septin ring formation along with Cla4 (Figure 3), as *bni1 cla4* double mutants form a cap of septins at the presumptive bud site that is not converted into a ring [151]. This could be related to the function of Cdc42 in polarized exocytosis along actin cables generated by Bni1 and in creating the hole in the middle of the septin ring [104].

Finally, Axl2 is a bud site landmark protein that also participates in septin recruitment (Figure 3) [163,164]. Recent two-hybrid and bimolecular fluorescence complementation (BiFC) assays indicated that Axl2 interacts with the septin Cdc10, as well as with Cdc42-GTP and Bud3, and contributes to the efficient gathering of septins to the cell division site [103]. 

In summary, while Cdc42 effectors, activators and cell polarity proteins have been implicated in septin organization, the precise mechanistic processes underlying septin recruitment and septin ring formation need to be further elucidated.

Another Rho GTPase, Rho1, has been implicated in timely and efficient septin recruitment and septin collar stability through activation of one of its downstream effectors, the kinase Pkc1 that in turn phosphorylates the septin-interacting protein Syp1 and promotes its turnover at the bud neck [165,166]. Syp1 appears at the presumptive bud site concomitant with septins and in vitro is able to align laterally and bundle septin filaments, thus offering mechanical stability to the new-forming septin ring/collar [167]. 

### 4.2. Maturation into the Septin Collar

As the bud emerges, the septin ring at the bud neck expands into a rigid hourglass-shaped septin collar. Whether this septin remodeling occurs suddenly at the G1/S transition or progressively throughout mitotic progression is not known. As mentioned above, the mature septin hourglass is a gauze-like, very stable septin assembly comprising axial and circumferential filaments (Figure 3) [49,84,85,92]. Likely, the transition from the initial septin ring, which is mainly composed by radial septin filaments converging towards a central hole [92], into the septin collar is controlled by the cell cycle machinery. 

One of the first septin-interacting proteins that was found to be involved in septin organization is the Nim-related kinase Gin4. Mutations in *GIN4* are synthetically lethal with *cdc12* and cause septins to form a fuzzy band or parallel axial “bars” at the bud neck instead of the hourglass [86]. A similar phenotype was observed in cells lacking the Gin4-binding protein Nap1 [14] or the septin Shs1 [13] and in cells exposed to mating pheromones [7,83,86]. These observations suggest that Gin4 may be involved in stabilising the circumferential septin filaments, which in turn require Shs1 for the overall gauze-like organization in a septin collar [92,94]. Consistently, Gin4 phosphorylates directly Shs1, thus contributing to the robustness of the hourglass [168,169]. Gin4 colocalises with septins from septin appearance throughout mitosis, but is displaced from the bud neck before septin ring splitting and is absent in septin structures of mating cells [86]. Its association with septins requires Nap1, Shs1, the PAK kinase Cla4 and cyclin-dependent kinases (CDKs) [168], thus partly explaining the links between septin collar organization and cell cycle progression. Accordingly, Shs1 phosphorylation by G1 CDKs stabilises the interaction of septins with Gin4 [170]. Two additional Nim-related kinases, Hsl1 and Kcc4, associate with septins in yeast cells [171,172,173,174,175,176,177]. Although the lack of Hsl1 or Kcc4 does not cause any obvious defect in septin organization [14], these kinases may share with Gin4 partially overlapping functions in building up the architecture of the septin collar [174]. 

During bud emergence, Gin4 targets to the bud neck the LKB1-like kinase Elm1, which contributes to septin collar architecture and stability until cytokinesis (Figure 3) [178,179,180,181]. In turn, Elm1 phosphorylates and activates Cla4, Gin4 and Hsl1 [14,168,169,180,181,182]. Its efficient recruitment to the bud neck requires the Dma1 and Dma2 ubiquitin ligases, which in turn are involved in proper septin stability at the bud neck through an unknown molecular mechanism [183,184]. The role of Elm1 in septin organization also involves the phosphorylation and functionality of Bni5 [179,185], a septin-interacting protein that was identified as a dosage suppressor of septin mutants [186]. Bni5 can dimerize and in vitro is able to crosslinks septin filaments into networks by bridging pairs or multiple filaments [185,187]. Additionally, Bni5 interacts with the C-terminal extension of Cdc11 and Shs1 and recruits Myo1 to the septin ring and collar throughout most of the cell cycle except during cytokinesis [110,188]. The exact role of Myo1 at the septin collar is unclear, but myosin II filaments may assemble on the membrane-distal side of the septin hourglass perpendicular to the axial septin filaments and somehow contribute to the appropriate septin architecture [92]. 

Post-translational modifications (PTMs) likely play an important role in the organization and stability of the septin hourglass. Besides the aforementioned septin phosphorylations by Cla4, Gin4 and CDKs, septins may be phosphorylated by other septin- and bud neck-associated kinases (e.g. Hsl1, Kcc4, polo kinase Cdc5, Mob1-Dbf2, etc.). Plenty of septin phosphorylation sites have been identified by phosphoproteomics, and some them are regulated across the cell cycle [189,190,191,192,193,194,195,196,197,198], suggesting that they may play an important role in the control of septin dynamics. However, their elevated number has so far hampered their systematic characterization. Septin SUMOylation occurs mainly during mitosis and could potentially promote the stability of the septin hourglass or septin interactions [199]. However, non-SUMOylatable septin mutants do not show any obvious defects in septin architecture and experience only a mild delay in septin disassembly after cell division [199]. Septin acetylation may also impact the overall architecture of the septin collar, as septin mutants with reduced acetylation levels have defects in septin localization [200]. Finally, some septins have also been found to be ubiquitylated [191], but the functional significance of this PTM has not been explored so far.

Since the septin collar is necessary for the recruitment of several cytokinetic proteins to the bud neck (see above), it is often considered as a “scaffold” for cytokinesis. However, septins actually inhibit AMR constriction and are no longer required for cytokinesis after AMR assembly [96,143]. Thus, whether the term “scaffold” is appropriate may be questionable. Why certain cytokinetic factors require to bind the septin collar in order to be incorporated into the AMR remains to be further investigated. Possibly, the septin collar congregates these proteins at the bud neck to reach threshold concentrations necessary for AMR assembly, thus acting as a molecular crowder. A non-mutually exclusive hypothesis is that binding to septins may induce specific protein conformations that are compatible with AMR interactions. 

### 4.3. Transition into the Septin Double Ring

During cytokinesis, the septin collar undergoes a sudden and dramatic reorganization, resulting in its splitting into two distinct rings that sandwich the AMR (Figure 3) [105,141]. This septin displacement is crucial for AMR constriction and cytokinesis, indicating that building a force-generating AMR requires at least two septin-related steps. Initially, the septin collar spurs the assembly of the cytokinetic machinery at the right place while preventing AMR-driven membrane ingression; subsequently, the confined eviction of septins from the division site during septin ring splitting initiates AMR constriction [96,142]. 

Formation and maintenance of the double septin ring depend on the anillin homologue Bud4 and the Rho-GEF Bud3 (Figure 3) [18,94,201,202,203], which have been previously implicated in the axial budding pattern of haploid yeast cells [101,202,204]. In the absence of Bud3 and Bud4 the septin collar disassembles completely at mitotic exit, without any obvious consequence on the kinetics of AMR constriction and cytokinesis [201], in line with the notion that septin clearance after AMR assembly is not detrimental for cytokinesis. Both Bud3 and Bud4 localise as a double ring at the edges of the septin collar during mitosis and then remain associated with the split septin rings, suggesting that these proteins act as spatial cues to pre-pattern the septin double ring [100,101,201,205,206,207]. 

Septin ring splitting requires the activity of the mitotic exit network (MEN; Figure 3), a Hippo-like pathway triggered by a top GTPase (Tem1) that turns on two sequentially acting protein kinases (Cdc15 and Mob1-Dbf2) to ultimately promote the activation of the Cdc14 phosphatase (reviewed in [208]). In turn, Cdc14 dephosphorylates and activates its upstream kinases Cdc15 and Mob1-Dbf2 in a positive feedback loop [209,210]. Cdc14 is the main CDK-counteracting phosphatase in budding yeast, and its function, together with that of its upstream MEN regulators, is essential for mitotic exit and cytokinesis [211,212,213,214,215,216]. Mitotic exit elicits several processes, such as spindle disassembly and licensing of replication origins, and it is an essential prerequisite for cytokinesis through dephosphorylation of specific mitotic CDK substrates [217]. However, MEN also promotes septin ring splitting independently of its role in mitotic exit [96], suggesting that one or more MEN targets prompt septin remodelling at cytokinesis. We have recently identified the cytokinetic protein Hof1 as one of such septin ring splitting regulators (Figure 3) [218]. Hof1 is involved in cytokinesis and is an established target of MEN [113,128,219,220]. It plays a major role in the control of actin polymerization and bundling, partly by modulating formin activity [221,222,223,224,225,226]. Consistently, it has been implicated in polarised growth [224,227]. However, the cytokinetic function of Hof1 was mainly linked to activation of the chitin synthase Chs2, and *CHS2* overexpression or gain of function mutations can suppress the lethality of *hof1*Δ mutants at high temperatures [113,127]. Our recent data indicate that *hof1* mutants also display defects in septin remodeling at cytokinesis [218]. During mitosis, Hof1 associates with septins, forming two closely spaced rings at the edges of the septin collar [128,218,219,228]. In vitro, it can induce the formation of intertwined septin bundles, suggesting that during this cell cycle stage it may contribute to the robustness of the septin hourglass [218,225]. However, shortly before septin ring splitting, Hof1 is displaced from the septin collar and relocates to the AMR, where it partially constricts alongside it [128,218,219,228]. This rapid translocation from septins to the AMR is triggered by Hof1 phosphorylation, primarily by the MEN kinase Dbf2, which disengages Hof1 from the septins, allowing it to join the AMR [218,219,220]. Phospho-mimicking *HOF1* mutant alleles can bypass the septin reorganization defects seen in MEN mutants by displacing Hof1 from septins and enhancing its translocation to the AMR. Importantly, septin remodeling by Hof1 depends on its membrane-binding F-BAR domain, suggesting that a local membrane reorganization could underlie septin disassembly from the cleavage site and remodelling into the double ring [218]. Since BAR domains can induce membrane curvature [229,230], Hof1 may bend the membrane at the division site to a radius that causes septin disassembly. Another possibility, stemming from the ability of BAR domains to cluster phosphoinositides [229,230], is that Hof1 may promote septin ring splitting by modifying the local composition of the plasma membrane. The non-essentiality of *HOF1*, especially in some strain background, suggests that additional, as-yet unidentified proteins participate in septin ring splitting alongside Hof1.

## 5. Conclusions and Perspectives

Cytokinesis is a fundamental yet vulnerable process that requires the orchestration of many players. Given the prominent role of septins in budding yeast cytokinesis, understanding how septin remodeling is controlled during the yeast cell cycle is key in order to gain insights into how cytokinesis is regulated in time and space. Unresolved questions that deserve further investigation regard the precise mechanism underlying septin membrane interactions, the impact of septin-binding proteins and post-translational modifications on septin structural organization, and the possible interplay between septins and other cytoskeletal elements. Unravelling how the yeast septin collar inhibits AMR constriction is another critical question for future research. As we continue to tease apart the complexity of septin biology, their integral role in cellular organization and function becomes increasingly apparent, opening exciting avenues for our comprehension of septin functions in physiological and pathological conditions.

## Figures and Tables

**Figure 1 jof-10-00642-f001:**
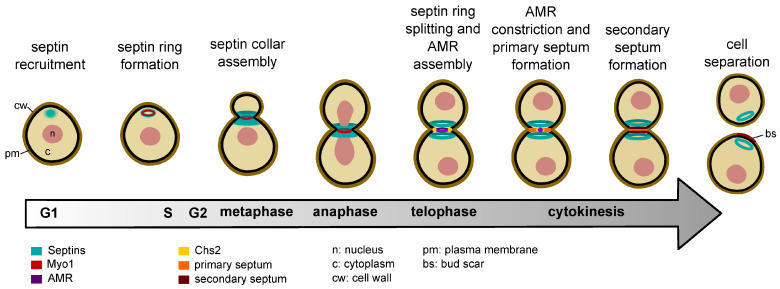
Main cell cycle events relevant for budding yeast cytokinesis and cell division. See text for details.

**Figure 2 jof-10-00642-f002:**
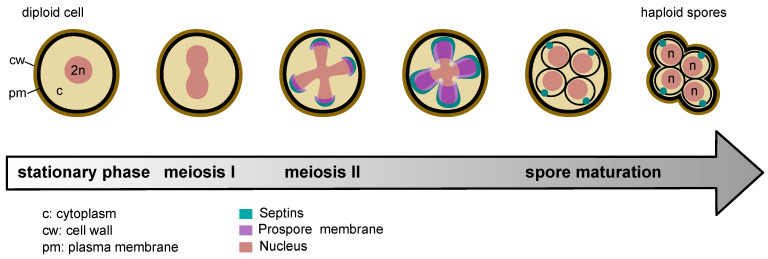
Septin localization during meiosis in budding yeast. Upon nitrogen deprivation and in the presence of poor carbon sources, budding yeast diploid cells stop dividing (stationary phase) and undergo meiosis. See text for details.

**Figure 3 jof-10-00642-f003:**
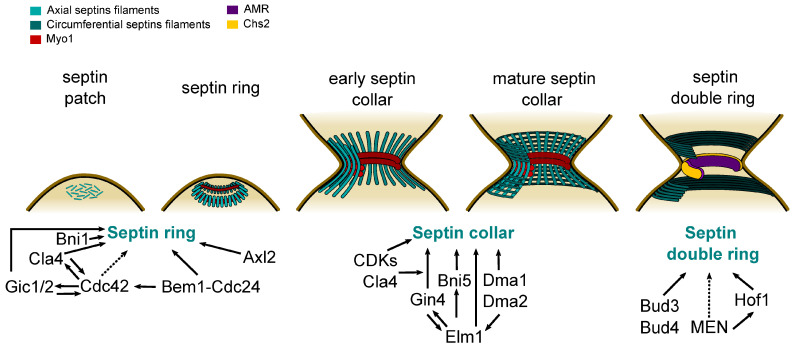
Septin architecture during different cell cycle phases. Septins are first recruited in G1 to the presumptive bud side as an unorganized patch that is rapidly converted into a cortical ring formed by radial septin filaments. Septin recruitment and/or ring formation involve the Cdc42 GTPase, in turn activated by the scaffold Bem1 and the GEF Cdc24, the Cdc42 effectors Cla4 and Gic1-2, the formin Bni1 and the axial landmark protein Axl2. At S phase entry, the septin ring expands into a septin collar that spans the bud neck and then matures into an hourglass-shaped structure formed by axial and circumferential septin filaments. Septin collar assembly involves the kinases Cla4, Gin4 and Elm1, CDKs, the septin-interacting protein Bni5 and the ubiquitin ligases Dma1 and Dma2. At the onset of cytokinesis, the axial septin filaments are depolymerized and possibly repolymerized into two arrays of circumferential filaments that make a double ring sandwiching the constricting AMR. Septin reorganization requires the MEN (mitotic exit network), its phosphorylation target Hof1 and other unidentified effectors. The septin double ring is patterned and stabilised by the Bud3 protein and the anillin-like Bud4. Solid arrows indicate established regulatory relationships; dashed arrows indicate hypothetical regulatory interactions. See text for further details.

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
