# Peer review of "Septin Organization and Dynamics for Budding Yeast Cytokinesis"

_jof, 2024, doi:10.3390/jof10090642_

Round 1

Reviewer 1 Report

This is an excellent and timely review of this topic. It will be of great value to the field and I look forward to sharing it with the members of my lab. The writing is crisp and clear and the figures are very informative and easy to understand. I have only four minor points.

Lines 121-125: “prospore walls” should be “prospore membranes”. After the last sentence in this paragraph, it would be worth adding mention of the observation that in fact there is in fact some particularly interesting non-uniformity of septin localization around the spore membrane: Cdc10 (the only septin carefully analyzed so far) localizes to a single punctum per spore generally located on the cortex away from the sites of contact between spores. Furthermore, this punctum marks the location where the spore will polarize upon germination. Reference PMIDs 17999778 and 33238051.

Line 178: “by recruiting to the bud neck landmarks proteins, such as Bud3 and Bud4”: Bud3 and Bud4 are not themselves consistently considered to be “landmarks”, since they lack transmembrane or extracellular domains, and instead are thought to link peripheral membrane proteins like septins to the glycosylated, transmembrane landmarks like Bud8, Bud9, Rax1, Rax2, and Axl2. I don’t think it’s worth going in too much detail here but it is probably worth clarifying the wording so as not to be unclear.

With regard to “scaffold” being an appropriate term: I see the authors’ point, though I think the term has come to be used in molecular biology in a way that is consistent with congregating proteins to reach threshold concentrations. In my opinion, it’s not functionally misleading to continue to use “scaffold”. But that is just my opinion.

This review is both focused and extensive and I like the range of topics it currently includes. The only topic I think would be worth including is the “morphogenesis checkpoint”: the cell cycle delay that occurs when a septin ring is not “properly” assembled and fails to recruit Hsl7 in a “proper” way. This phenomenon suggests that cells have evolved rather elaborate ways to “monitor” the state of higher-order septin assembly, but there are still many interesting open questions in terms of exactly what is being monitored.

Author Response

1) Lines 121-125: “prospore walls” should be “prospore membranes”. After the last sentence in this paragraph, it would be worth adding mention of the observation that in fact there is in fact some particularly interesting non-uniformity of septin localization around the spore membrane: Cdc10 (the only septin carefully analyzed so far) localizes to a single punctum per spore generally located on the cortex away from the sites of contact between spores. Furthermore, this punctum marks the location where the spore will polarize upon germination. Reference PMIDs 17999778 and 33238051.

Our answer: Thank you for the suggestion, we have now extended the text to refer to these interesting papers. We also changed “prospore walls” into “prospore membranes”.

2) Line 178: “by recruiting to the bud neck landmarks proteins, such as Bud3 and Bud4”: Bud3 and Bud4 are not themselves consistently considered to be “landmarks”, since they lack transmembrane or extracellular domains, and instead are thought to link peripheral membrane proteins like septins to the glycosylated, transmembrane landmarks like Bud8, Bud9, Rax1, Rax2, and Axl2. I don’t think it’s worth going in too much detail here but it is probably worth clarifying the wording so as not to be unclear.

Our answer: Thank you for this clarification. With the term “landmark proteins” we refer to all proteins that mark the future bud site, independent of whether they bind to the plasma membrane. We have  slightly modified the text (line 159) to clarify this point.

3) With regard to “scaffold” being an appropriate term: I see the authors’ point, though I think the term has come to be used in molecular biology in a way that is consistent with congregating proteins to reach threshold concentrations. In my opinion, it’s not functionally misleading to continue to use “scaffold”. But that is just my opinion.

Our answer: We have now toned down our argument (lines 331-332) to leave open the possibility that septins do act as a scaffold.

4) This review is both focused and extensive and I like the range of topics it currently includes. The only topic I think would be worth including is the “morphogenesis checkpoint”: the cell cycle delay that occurs when a septin ring is not “properly” assembled and fails to recruit Hsl7 in a “proper” way. This phenomenon suggests that cells have evolved rather elaborate ways to “monitor” the state of higher-order septin assembly, but there are still many interesting open questions in terms of exactly what is being monitored.

Our answer: We initially hesitated about whether to include the morphogenesis checkpoint in this Review. However, what the morphogenesis checkpoint actually monitors is still an open question, as pointed out by this Reviewer. Data from the Kellogg’s lab (Anastasia et al., 2012, DOI: 10.1083/jcb.201108108 ) led to the proposal that plasma membrane growth/composition/architecture may be sensed by the checkpoint to eventually delay mitotic entry in case of defects. Thus, septin mis-assembly may indirectly activate the morphogenesis checkpoint by impacting membrane organization. For this reason, we decided not to include a section on this topic, as we felt it may have been lengthy and divert the reader’s attention from the main focus of the Review.

Reviewer 2 Report

In this manuscript, the authors review the control of septin organization during the cell cycle in the budding yeast Saccharomyces cerevisiae. In Saccharomyces cerevisiae, septins play a pivotal role in orchestrating cytokinesis by promoting the assembly of the cytokinetic machine at the division site and septins are also required to control the activity of the machine. The authors did a good job in summarizing the functions of yeast septins during cytokinesis, discussing the regulation of septins, and making sound implications the septin findings.

Overall, the work covers a large number of studies on budding yeast septins and deserves to be published. The following points may be considered for improvements.

Major Points

1)      Based on the title “The control of septin dynamics for budding yeast cytokinesis”, I expect discussions on the control of the stability/instability of septins. However, the author seems to focus on “the organization of septins during the cell cycle”, and the dynamics of septin organization during the cell cycle. I wonder whether the title should be modified to better reflect the content.

2)      Septins play an important role in regulating meiosis in yeasts. Perhaps, the author should add a graph to illustrate the localization and organization of septins during meiosis (L 117-125).

3)      Septins also play key roles during the “separation stage” both in mammalian cells and fission yeasts. Why this function is absent in budding yeast (L 229-235)? The authors may discuss this point.

Minor point

L 369, the subtitle is “2 Transition into the septin double ring”.

see the report above

Author Response

Major Points

1)      Based on the title “The control of septin dynamics for budding yeast cytokinesis”, I expect discussions on the control of the stability/instability of septins. However, the author seems to focus on “the organization of septins during the cell cycle”, and the dynamics of septin organization during the cell cycle. I wonder whether the title should be modified to better reflect the content.

Our answer: We have slightly changed the title into "Septin organization and dynamics for budding yeast cytokinesis". We feel it is important to have "cytokinesis" in the title since the Review will be part of a special JoF issue on Cytokinesis. We also think that, given that yeast septins play a major role in cytokinesis, their organization and dynamics even at early cell cycle stages is important for this process. 

2)      Septins play an important role in regulating meiosis in yeasts. Perhaps, the author should add a graph to illustrate the localization and organization of septins during meiosis (L 117-125).

Our answer: Thank you for the suggestion. We have now included a new figure (Fig. 2) to illustrate septin localization in meiosis. We have also slightly extended the related main text. 

3)      Septins also play key roles during the “separation stage” both in mammalian cells and fission yeasts. Why this function is absent in budding yeast (L 229-235)? The authors may discuss this point.

Our answer: As we argue in the text, whether septins are also involved in cell separation in budding yeast is not known. Their prominent involvement in cytokinesis hampers the possibility to address this question. The only way to address this issue would be to generate separation of function septin mutants, or to bypass the cytokinesis defects of existing septin mutants. 

Minor point

L 369, the subtitle is “2 Transition into the septin double ring”.

Our answer: thank you for noticing, the error came from the JoF formatting where the numbering of the subtitles has disappeared for some paragraphs.